# Impact of Extreme Temperatures on Ambulance Dispatches Due to Cardiovascular Causes in North-West Spain

**DOI:** 10.3390/ijerph17239001

**Published:** 2020-12-03

**Authors:** Santiago Gestal Romani, Dominic Royé, Luis Sánchez Santos, Adolfo Figueiras

**Affiliations:** 1Hospital of Montecelo Pontevedra, 36071 Pontevedra, Spain; 2Department of Preventive Medicine and Public Health, University of Santiago de Compostela, 15782 Santiago de Compostela, Spain; dominic.roye@usc.es (D.R.); adolfo.figueiras@usc.es (A.F.); 3Consortium for Biomedical Research in Epidemiology and Public Health Network, (CIBERESP), 15782 Santiago de Compostela, Spain; 4Health Research Institute of Santiago de Compostela (IDIS), 15782 Santiago de Compostela, Spain; 5Galician Public Health Emergencies Foundation-061, 36680 A Estrada, Spain; Luis.Sanchez.Santos@sergas.es

**Keywords:** ambulance dispatches, extreme temperature, Galicia, cardiovascular diseases, quasi-Poisson regression model

## Abstract

**Introduction and objectives**. The increase in mortality and hospital admissions associated with high and low temperatures is well established. However, less is known about the influence of extreme ambient temperature conditions on cardiovascular ambulance dispatches. This study seeks to evaluate the effects of minimum and maximum daily temperatures on cardiovascular morbidity in the cities of Vigo and A Coruña in North-West Spain, using emergency medical calls during the period 2005–2017. **Methods.** For the purposes of analysis, we employed a quasi-Poisson time series regression model, within a distributed non-linear lag model by exposure variable and city. The relative risks of cold- and heat-related calls were estimated for each city and temperature model. **Results.** A total of 70,537 calls were evaluated, most of which were associated with low maximum and minimum temperatures on cold days in both cities. At maximum temperatures, significant cold-related effects were observed at lags of 3–6 days in Vigo and 5–11 days in A Coruña. At minimum temperatures, cold-related effects registered a similar pattern in both cities, with significant relative risks at lags of 4 to 12 days in A Coruña. Heat-related effects did not display a clearly significant pattern. **Conclusions.** An increase in cardiovascular morbidity is observed with moderately low temperatures without extremes being required to establish an effect. Public health prevention plans and warning systems should consider including moderate temperature range in the prevention of cardiovascular morbidity.

## 1. Introduction

The importance currently attached to global climate change has aroused great interest in studying the relationships between climate and health. One of the climatic factors which has caused greatest concern is temperature. The relationship between temperature and mortality is well established, with a J, V or U-shaped relationship, starting from a minimum mortality temperature (lower risk) and increasing as temperatures rise or fall [1,2]. It has also been observed that the minimum mortality temperature and the functional exposure-response curve vary from one place to another due to climatic conditions and the resilience of the population, among other factors [3].

Cardiovascular diseases are the leading cause of death in developed countries, with their incidence being associated with ambient temperature [3]. On days with extreme temperature, an increase has been observed in the risk of acute myocardial infarctions [4] and cerebrovascular accidents [5]. Likewise, the incidence of episodes of paroxysmal atrial fibrillation, which can cause a decompensation and increase cardiovascular morbidity, has also been associated with temperature [6]. Under certain hot and cold environmental conditions, physiological body temperature regulation mechanisms may become incapable of maintaining thermal equilibrium, leading to alterations that place considerable stress on the cardiovascular system [7].

Many studies published to date have employed outcomes such as mortality, hospital admissions and emergency department data [8,9,10]. Moreover, a contrasting pattern of temperature-related mortality and hospitalisation can be observed [11].

These types of study only record the most severe cases and, thus, only cover a limited proportion of patients with cardiovascular problems. These variables may not be sufficiently sensitive to moderate environmental changes [12,13]. Calls to emergency medical services due to cardiovascular causes may better reflect the influence of ambient temperature on the incidence of cardiovascular morbidity as they are capable of registering small decompensations not requiring hospital care. To date, very few studies have examined the use of emergency medical calls due to cardiovascular causes and their relationship with temperature.

We sought to study the relationship between extreme daily (minimum and maximum) temperatures and emergency medical service calls due to cardiovascular causes in two medium-sized cities in North-West Spain (Vigo and A Coruña) with a predominantly oceanic climate.

## 2. Methods

### 2.1. Settings

This study was carried out in two cities (A Coruña and Vigo) located on the Atlantic coast of Galicia (North-West Spain). A Coruña (latitude 43.3°, longitude −8.4°) has a population of 245,711 inhabitants, 24.5% of whom are over 65 years of age, while Vigo (latitude 42.2°, longitude −8.7°) has a population of 295,364, with 22.4% over 65 years of age. A Coruña has a southern oceanic climate and temperatures remain mild throughout the year, with a low annual temperature range. Vigo’s climate is one of transition between the Mediterranean oceanic climate and the oceanic climate, with high rainfall.

The Galicia-061 Public Health Emergency Foundation—Fully Integrated Emergency Care Centre (Fundación Pública Urxencias Sanitarias de Galicia-061—Centro Integrado de Atención ás Emerxencias) is a public and universal health care system which manages the demand for urgent healthcare, offers medical advice via telephone and mobilises the necessary health resources (ambulances) to be sent to the scene of medical incidents.

### 2.2. Healthcare Data

Emergency medical calls were obtained from the Galicia-061 Public Health Emergency Foundation—Fully Integrated Emergency Care Centre for the period 1 January 2005 to 14 November 2017 in the cities of Vigo and A Coruña. The medical alert date was used as a time reference. The database was filtered by cardiovascular cause (International Classification of Diseases, 9th and 10th revisions; ICD-9: 390–459, ICD-10: I00-I99), based on three variables with a diagnostic record of the alert. The three diagnostic records corresponded to: the suspected diagnosis by central emergency care staff; the diagnosis by medicalised-ambulance staff; and the diagnosis by primary care staff. We included all alerts registered with at least one diagnosis of cardiovascular causes in one of the three variables. No distinction was drawn concerning the nature of termination of the alert (death, transfer to hospital, etc.).

### 2.3. Temperature Data

Data on minimum (Tmin) and maximum daily temperatures (Tmax) recorded by the meteorological stations located at Vigo airport and in the city centre of A Coruña were obtained from the State Meteorological Agency (*Agencia Estatal de Meteorología*), for the study period (1 January 2005 to 14 November 2017).

### 2.4. Statistical Analysis

The associations between emergency medical calls due to cardiovascular causes and maximum and minimum temperature were modelled using a Quasi-Poisson time series regression model within a non-linear lag model distributed according to each exposure variable and city [14,15]. A distributed lag non-linear model (dlnm) simultaneously fits complex non-linear and lagged effects of an environmental variable on a response variable in generalised linear models or generalised additive models. The statistical details of the model are described by Bhaskaran et al. [16] and Gasparrini et al. [17].

To check for long-term trend and seasonality, we included a natural cubic spline with 7 degrees of freedom (df) per year [17]. In addition, adjustment for the possible confounding effects of the day of the week was made with an indicator. The lag response curve for the variable of exposure was modelled using a natural cubic B-spline and three interior knots (equally-spaced log values of lags). For the relationship between temperature and emergency call, we used a cross-basis matrix consisting of a quadratic B-spline with three interior knots at specific temperature percentiles for each city (10%, 75% and 90%). The lag period was extended to 14 days to capture the lag in cold-related effects [17]. The threshold temperature (the temperature at which the relative risk of calls is minimal), used as a reference value was estimated using the method applied by Tobías et al. [18].

For each city and temperature model, we estimated the overall cumulative relative risks of calls associated with cold and heat, respectively. This was defined as the risk of an increase in the 5% and 95% percentiles of each temperature distribution in relation to minimum morbidity temperature (MMT).

The risk of mortality attributable to a temperature *x_t_* for a given day *t* in the time series is defined as the number *AN_x,t_* and fraction *AF_x,t_* of calls registered in the next *L* days, with *L* being the maximum lag period, defined by:$$AF_{x,t}=1-exp\left(-\overset{L}{\underset{l=0}{\sum }}\beta_{x_{t},l}\right)$$
and
$$NA_{x,t}=\;AF_{x,t}\cdot \;\overset{L}{\underset{l=0}{\sum }}\frac{n_{t+1}}{L+1}$$
where $\sum^{\;}\beta_{x_{t},l}$ is the overall log-cumulative risk of temperature *x_t_* on day *t*, and *n_t_* is the number of calls on day *t*. This method is described in detail by Gasparrini and Leone [19].

In the sensitivity analysis, different parameterisations were tested, checking for various degrees of freedom for the variables (3 to 6 df) of seasonality and trend (6 to 10 df per year). All statistical analyses were performed with the R software system (version 3.6.2), using the mgcv (version 1.8-15) and dlnm packages (version 2.2.6) [20].

## 3. Results

Throughout the study period (1 January 2005 to 14 November 2017), 70,537 calls were made to the Galician emergency healthcare system due to cardiovascular causes (37,278 in Vigo and 33,259 in A Coruña). Figure 1 shows the trend in the daily series of these calls from 2005 to 2017, showing a similar seasonal variability for both cities.

The distribution according to sex in the two cities displayed the same pattern, with 59% of emergency call patients being women and only 40.8% men; 76.2% of all patients were over 64 years of age. The most frequently diagnosed diseases were stroke, cardiac failure, and chest pain of an ischaemic nature.

The Tmax and Tmin values in Vigo and A Coruña were similar, with total calls and maximum values being higher in Vigo (Table 1, Appendix A).

The MMT, at which the effect of temperature on calls is estimated to be minimal, with a relative risk (RR) of 1.0, was observed at maximum and minimum temperatures of 28.3 °C (95% confidence interval (CI), 4.5–39.5 °C) and 16.5 °C (95% CI, 15.0–22.5 °C) in A Coruña and 28 °C (95% CI, 14.1–40.8 °C) and 15.4 °C (95% CI, −2.7–16.7 °C) in Vigo (Figure 1). The impacts of extreme temperatures on the RR of morbidity are summarised in Figure 1. The correlation between emergency calls and extreme temperatures was positive, with a non-linear U- and J-shaped increase in effects observed at high temperatures in the two cities studied. Cold-related effects were more clearly observable compared to those estimated for heat.

Figure 2 shows the lags of the overall cumulative heat- and cold-related effects of Tmax and Tmin on emergency medical calls due to cardiovascular diseases in A Coruña and Vigo. In the case of the effects of Tmax, there was a 2-day lag in the appearance of cold-related effects in Vigo, with a significant RR after lags of 3 to 6 days and 5 to 11 days in A Coruña. The highest RRs were observed as follows: 1.06 (95% CI, 1.03–1.09) at a lag of 3 to 4 days in Vigo, and 1.03 (95% CI, 1.01–1.05) at a lag of 5 to 6 days in A Coruña due to the effects of cold. Estimates of cold-induced effects were higher for Vigo than for A Coruña. Regarding the possible effects attributable to heat in Tmax, no significant RRs were found.

At Tmin, cold-related effects displayed a similar pattern, with significant RRs at lags of 4 to 12 days in A Coruña, rising to a maximum of 1.03 (95% CI, 1.02–1.05) at a lag of 6 days. Cold-related effects in Vigo remained similar, although non-significant, in all the lags, with an RR of 1.01 being observed (95% CI, 0.99–1.03) at a lag of 6 days.

Possible heat-related effects showed no clearly significant pattern. However, mention should be made of the heat-induced effect in the city of Vigo, with an RR of 1.01 (95% CI, 0.99–1.02) at a lag of 1 day. This is extremely clear in Figure 2B at a high minimum temperature. In contrast, no heat-related effects were detected in A Coruña.

Table 2 shows excess emergency calls attributable to heat and cold, according to the exposure variable and city. Cold-related cardiovascular morbidity at maximum temperatures was estimated at 14% (95% CI, 5.4–21.6%) and 7.4% (95% CI, −1.8–15.3%) for A Coruña and Vigo, respectively. This fraction translated as 4650 excess annual calls (95% CI, 1782–7199) for A Coruña and 2751 (95% CI, −659–5710) for Vigo. Comparison between the cities showed that excess calls due to cold temperatures were much higher in the case of A Coruña. The observed effects showed a clear association between medical calls and low temperatures of maximum and minimum exposure. In contrast, excesses due to heat were manifestly lower in magnitude, with 0.3% (95% CI, −0.3–0.8%) and 0.4% (95% CI, −0.2–0.9%) for exposure to Tmax in A Coruña and Vigo, respectively.

## 4. Discussion

This study examines the effects of extreme temperatures on emergency medical calls due to cardiovascular issues in two medium-sized cities (Vigo and A Coruña) in North-West Spain with an oceanic climate. The results suggest that the number of daily emergency calls are associated with lower minimum and maximum temperatures. These findings could indicate that low temperatures are associated with more cases of emergency calls than high temperatures, at least in cities with an oceanic climate.

The effect found of moderate temperatures on ambulance dispatches due to cardiovascular causes is consistent with that found in a nationwide study in Japan [21]. In that study, it was observed that extreme temperatures have less impact [attributable fraction (AF) = 0.1%] on the number of calls than moderate temperatures, which are responsible for 18% of calls for cardiovascular causes. A similar AF was found for A Coruña (15.7%). The results obtained from the emergency medical system in Hamburg (Germany) [22] also demonstrated this greater impact for moderate temperatures. The most plausible explanation is that the frequency of days with moderately low or high temperatures is higher than that of days with extreme temperatures. As far as subtypes of cardiovascular events are concerned, greater effects are shown to be induced by low rather than by high temperatures [23,24,25,26,27,28,29,30].

Our results contrast with those of other studies that have also evaluated the impact of temperature on ambulance callouts, specifically in London [31], and in Luoyang (central China) [32], which found a relationship between ambulance calls for heat and cold. However, the exposure-response curve was clearly dependent on the dispatch category [31].

Not all studies use the same measure of temperature to evaluate effects on health. In our case, we opted to study the effects of extreme daily temperature (Tmax and Tmin) due to the fact that this, as compared to mean temperature, is a measure with which less exposure data are lost, given that the mean temperatures are not capable of reflecting daily extremes [33]. Other studies [34,35] have evaluated the effect via the apparent temperature or variation in daily temperature, which are other types of independent effects on temperature-related morbidity.

The observed MMT was estimated at 15.4 °C in Vigo and 16.5 °C in A Coruña for the minimum and at 28 °C in Vigo and 28.3 °C in A Coruña for the maximum temperature. The small difference in the MMT between the two cities can be accounted for by the difference in the location of the meteorological stations. It may also be a reflection of the mild climate in both cities in terms of low temperatures. Similar results were reported in Adelaide [36] and Brisbane [37].

Our study found a U- or J-shaped relationship between temperature and emergency calls due to cardiovascular causes, similar to that observed in previous studies [34,35,36,37,38]. The effects are not very pronounced for moderate temperatures, although they grow as they move towards the extremes (Figure 1) [2,3].

In terms of lag structure, the appearance of the effect at low temperatures is found in lags of 3–6 days in Vigo and 4–11 days in A Coruña, and at high temperatures in lags of 1–2 days in Vigo. These results are comparable to those of studies in which temperature and morbidity are correlated with hospital admissions or emergency care episodes [8,9], where the effects due to high temperatures are observed on the same day or the following day. These studies show that the effects due to low temperatures tend to appear two days after exposure and are then seen to persist for 2 to 3 days. In mortality studies, by contrast, the effects induced by low temperatures are observed over a longer period [39].

The pathophysiological mechanism involved in the increase in cardiovascular events on days with low temperatures is not clear. It would appear that in the case of susceptible individuals, changes induced at the different levels described below could precipitate cardiovascular decompensation [40]. At a cutaneous level, exposure to cold stimulates the thermo-receptors, producing catecholamine secretion and vasoconstriction [41], and increases heart rate, blood pressure and oxygen demand. At a renal level, diuresis is increased and, with plasma volumes depleted, blood viscosity increases [40]. At a pulmonary level, mucociliary activity decreases, with the possibility of causing bronchoconstriction, the release of biochemical mediators by lung cells, and a tendency to develop infections. In terms of blood, there is an increase in acute phase reactants, platelet aggregability and cholesterol values, all of which are cardiovascular risk factors [41].

A heat-related effect with a high Tmin was observed in the city of Vigo, which was close to statistical significance, RR = 1.01 (95% CI, 0.99–1.02) at lag 1 (Figure 2B. This is an effect which has not been seen in many studies and may be related to the higher sensitivity of the measure used. Elevated night temperatures can precipitate cardiovascular decompensations in people with impaired thermoregulation mechanisms and trigger disturbances in adequate night rest (circadian rhythm) [41].

Our study has several advantages; the first is the use of ambulance calls as an indicator of cardiovascular morbidity. We believe that this is a more sensitive measure of morbidity since: (1) it covers a larger population and is more easily accessible than emergencies in healthcare centres, as no means of transport is required. This can be important in the case of older people who live alone or people with low income; (2) some pathologies can be treated in situ (hypertensive crises, mild heart failure symptoms, paroxysmal arrhythmias, dizziness in relation to hypotensive conditions, among others) and, therefore, are not registered as hospital emergencies. The second advantage of our study is that the estimation of the attributable fraction has allowed us to assess which temperature ranges have a greater impact on public health, thereby making it possible to (1) design interventions to mitigate exposure to these temperatures; and (2) adapt and optimise health services according to temperatures.

As far as possible *limitations* of this study are concerned, the following should be noted. Firstly, although well-established methods were used to control for trend and seasonality, it cannot be ruled out that the results obtained may have been affected by confounding temporal patterns due to unknown variables. Secondly, the diagnostic data used were those supplied by the Fully Integrated Emergency Care Centre and, although such data may vary after hospital care, thereby possibly detracting from the accuracy of the study, they nonetheless have the advantage of greater sensitivity when it comes to detecting less severe processes. Thirdly, the exposure assigned to the population was the exterior temperature, a fixed exposure which is applied equally to both cities, without taking into account intra-urban differences, individual acclimatisation, behavioural and socioeconomic factors or housing characteristics, all of which could modify exposure [39]. Fourthly, the fact of having a single monitoring station available in each city means that exposure data might not be sufficiently representative of the whole population. Furthermore, the effects obtained may be underestimated as the stations chosen might not include the influence of the urban heat island. That said, however, the location of the meteorological station at Vigo airport, at an altitude of approximately 200 metres, means that the population exposure data could be somewhat higher than the values obtained. While this might change the minimum morbidity value slightly, it would not change the overall effects due to thermal variations. Fifthly, possible confounding factors such as air pollution, which has been associated with episodes of cardiovascular morbidity [41], were not taken into account. However, there are studies which suggest that failure to adjust for environmental pollution does not alter results [42].

## 5. Conclusions

This is the first study conducted in North-West Spain to evaluate the effects of ambient temperature on cardiovascular morbidity using calls to the medical emergency system. Evidence was found to show that daily cardiovascular disease morbidity is associated with moderately low maximum and minimum temperatures.

Further research will be necessary to study deeper emergency call subgroups and other geographical areas as well as population subgroups. The results obtained should be included in morbidity-related prevention plans or warning systems and in ambulance forecast models. They should not only take into account extreme values but also moderately low or high temperatures, and, in addition, should be implemented at both a regional and a local level. This could prove highly relevant in the current context of population ageing and global warming.

## Figures and Tables

**Figure 1 ijerph-17-09001-f001:**
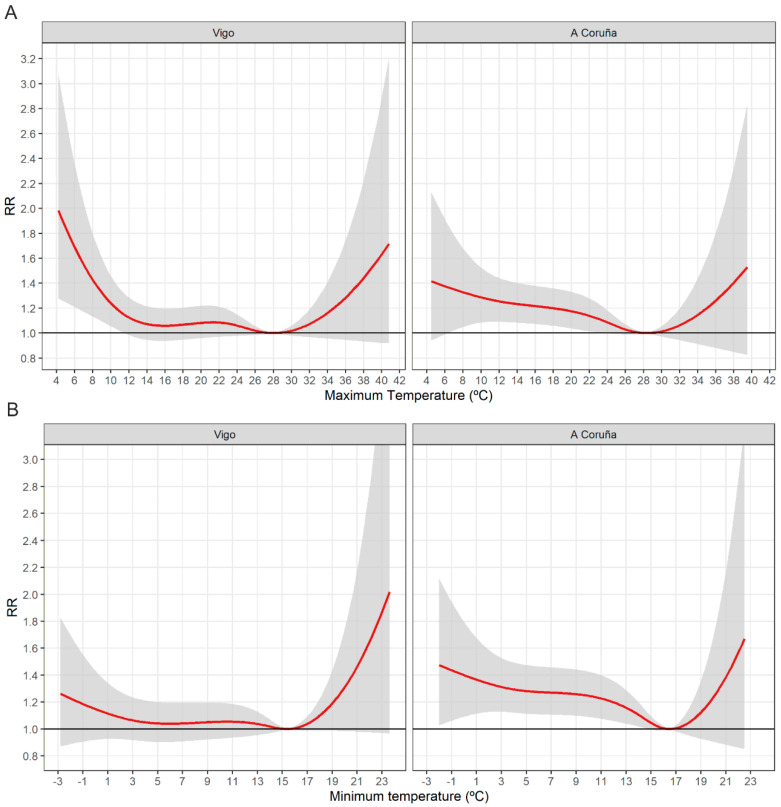
Overall cumulative exposure-response associations between maximum (**A**) and minimum temperature (**B**), and emergency medical calls in the cities of Vigo and A Coruña.

**Figure 2 ijerph-17-09001-f002:**
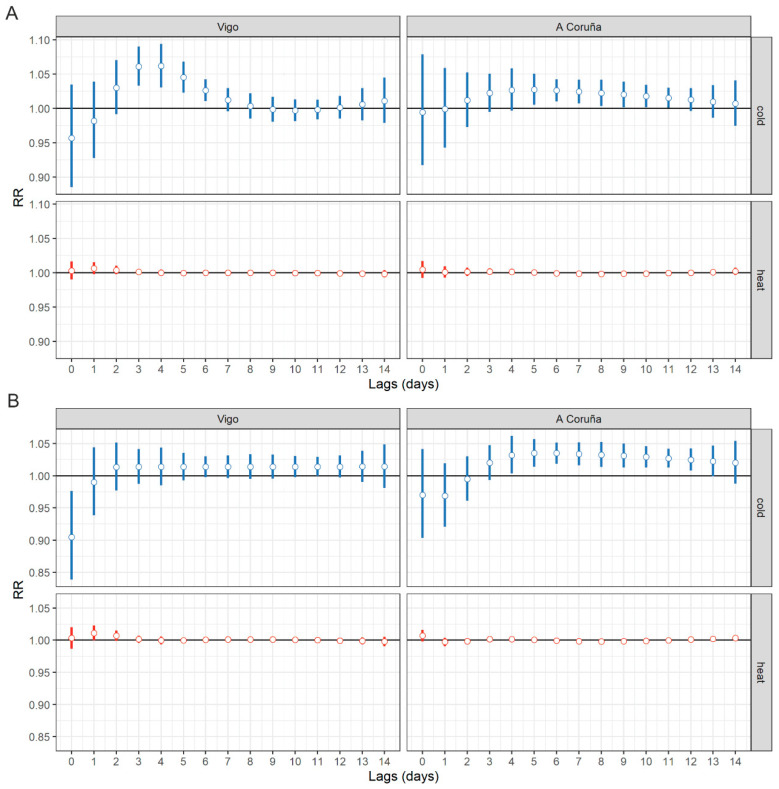
Structure of lags of overall cumulative heat-and cold-related effects of maximum (**A**) and minimum temperature (**B**) on emergency medical calls in the cities of Vigo and A Coruña.

**Table 1 ijerph-17-09001-t001:** Distribution of daily extreme temperature values and emergency calls (2005–2017).

	Maximum Temperatures (°C)			Minimum Temperatures (°C)			Calls (Number)		
City	Mean	Minimum	Maximum	Mean	Minimum	Maximum	Total	Mean	Maximum
**Vigo**	18.9	4.2	40.8	10.1	−2.8	22.5	37278	7.9	23
**A Coruña**	19.2	4.5	39.5	10.5	−2.0	23.6	33259	7.1	18

**Table 2 ijerph-17-09001-t002:** Emergency medical calls attributable to heat and cold by exposure variable and city.

City	Exposure	Effect	* AF% (95 CI%)	^‡^ AN (95 CI%)
**A Coruña**	Maximum	cold	14 (5.4. 21.6)	4650 (1782, 7199)
**A Coruña**	Maximum	heat	0.3 (−0.3. 0.8)	103 (−96, 270)
**Vigo**	Maximum	cold	7.4 (−1.8. 15.3)	2751 (−659, 5710)
**Vigo**	Maximum	heat	0.4 (−0.2. 0.9)	142 (−59, 319)
**A Coruña**	Minimum	cold	15.7 (7.3. 23.5)	5224 (2443, 7823)
**A Coruña**	Minimum	heat	0.3 (−0.2. 0.7)	89 (−81, 231)
**Vigo**	Minimum	cold	3.9 (−5.6. 12.5)	1448 (−2083, 4666)
**Vigo**	Minimum	heat	0.5 (−0.1. 0.9)	169 (−32, 339)

* AF: attributable fraction, ^‡^ AN: attributable number.

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
