# Peer review of "Impact of Extreme Temperatures on Ambulance Dispatches Due to Cardiovascular Causes in North-West Spain"

_ijerph, 2020, doi:10.3390/ijerph17239001_

Round 1

Reviewer 1 Report

I like the paper in principle but (i) I wonder what is really novel and (ii) the paper is really not very well written because sentences are constructed very oddly at times and the presentation is very limited so I needed to make a lot of effort to connect parts. The novelty cannot be just applying an already developed model on new dataset but about how results can inform about novel findings and I am not sure what these novel findings are. E.g. Liu et al (2018) found many things that the authors found as well as several other Gasparrini results.

One novelty can be the discussion about/if temperature specific threshold exist for the cities considered vs. others in order to minimize/maximize certain disease risk. This can be certainly novel as well as differences in relative risk curves.

Anyhow, I really suggest Major Revisions, mostly due to the highly fragmented and succinct writing that must improved with much more cohesion and substance.

Threshold Evaluation of Emergency Risk Communication for Health Risks Related to Hazardous Ambient Temperature

Yang Liu  Brenda O. Hoppe  Matteo Convertino

First published: 10 April 2018 https://doi.org/10.1111/risa.12998

Reviewer 2 Report

This paper evaluated the impact of daily minimum and maximum temperatures on cardiovascular morbidity using emergency medical calls across the period 2005-2017 in two cities of northwestern Spain. However, the following should be revised before further consideration:

  1. As described by the authors in conclusion, ambient temperature may have different effects in other geographic regions, the latitude and longitude of these two cities should be explained.
  2. The extreme temperatures in Table 1 are recommended to be expressed year by year.
  3. In Figure 1, there are still some differences in the minimum temperatures between the two cities. The authors are encouraged to explain whether the difference has been taken into consideration.
  4. In the section of RESULTS, please briefly describe the background of the software being used.
  5. In the analysis of gender, whether the caller and the patient is the same person, or whether the person who lives with him/her helps the call. Why is the age divided by 64 years? The authors are encouraged to explain further.
  6. Figures 3, 4, and 5 are not shown in this paper. The content after Line 137 cannot be commented.

Round 2

Reviewer 1 Report

I appreciate the comments of the authors and I believe the manuscript can be accepted at this stage. In the future I would look more into which set of environmental factors is more sensitive (and yet more indicative a priori) to certain morbidities/mortalities

M

Reviewer 2 Report

The authors have responded to my previous comments. I have no further comments.